DOI: 10.1038/s41467-018-04802-8　**OPEN**

# Continuous biomarker monitoring by particle mobility sensing with single molecule resolution

Emiel W.A. Visser[1,2], Junhong Yan[2,3], Leo J. van IJzendoorn[1,2] & Menno W.J. Prins [1,2,3]

Healthcare is in demand of technologies for real-time sensing in order to continuously guard the state of patients. Here we present biomarker-monitoring based on the sensing of particle mobility, a concept wherein particles are coupled to a substrate via a flexible molecular tether, with both the particles and substrate provided with affinity molecules for effectuating specific and reversible interactions. Single-molecular binding and unbinding events modulate the Brownian particle motion and the state changes are recorded using optical scattering microscopy. The technology is demonstrated with DNA and protein as model biomarkers, in buffer and in blood plasma, showing sensitivity to picomolar and nanomolar concentrations. The sensing principle is direct and self-contained, without consuming or producing any reactants. With its basis in reversible interactions and single-molecule resolution, we envisage that the presented technology will enable biosensors for continuous biomarker monitoring with high sensitivity, specificity, and accuracy.

[1] Department of Applied Physics, Eindhoven University of Technology, 5600 MB Eindhoven, The Netherlands. [2] Institute for Complex Molecular Systems (ICMS), Eindhoven University of Technology, 5600 MB Eindhoven, Netherlands. [3] Department of Biomedical Engineering, Eindhoven University of Technology, 5600 MB Eindhoven, Netherlands. These authors contributed equally: Emiel W.A. Visser, Junhong Yan. Correspondence and requests for materials should be addressed to M.W.J.P. (email: m.w.j.prins@tue.nl)

Sensing technologies for acquiring real-time, precise, and reliable data are becoming more and more important in the field of healthcare, to help monitor, treat and coach patients[1]. Over the past years, many sensors have become commercially available, e.g. for the sensing of heart rate, muscle action, electrocardiography (ECG), body temperature, blood pressure, and respiratory rate[2]. These sensors all measure physical properties rather than underlying biochemical processes. An important next step is to develop sensors for the continuous in-situ monitoring of chemical and biochemical markers. Devices for continuous glucose monitoring are commercially available[3] and are actively being studied[4–6], but the underlying enzyme-based electrochemical-sensing principle is not suitable for other important biomarkers, such as hormones, drugs, peptides, proteins, and nucleic acids. Such sensors are needed, e.g. for optimal patient treatment by monitoring of therapeutic drug levels[7,8], for patient safety by measuring organ failure markers[9,10], and for coaching of patients by monitoring stress markers[11].

Arroyo et al.[7] demonstrated the monitoring of small-molecule therapeutic drugs in live mice using electrochemical sensing based on aptamer-folding, with sensitivities in the low-micromolar range. The sensing principle is based on molecular probes that change conformation upon binding or unbinding of target molecules, thereby modulating the proximity of a redox-active group to an electrode surface. The sensing principle is very elegant and biomedically relevant results have been demonstrated. Unfortunately libraries of existing affinity binders, such as antibodies cannot be easily implemented in the system, due to the requirement of conformational switching. Furthermore, there is an interest in the field to be able to measure at even lower concentrations, since many biomarkers are present at nanomolar and picomolar levels.

To achieve high sensitivity and general applicability, an ideal biosensing technique should be compatible with a wide variety of affinity binders and should enable single-molecule resolution. Achieving single-molecule resolution gives the opportunity to enhance sensitivity, specificity, precision, and accuracy. For academic studies a range of single-molecule techniques have been developed[12], such as fluorescence resonance energy transfer[13], tethered particle motion[14,15], nanopores[16], and super-resolution microscopy[17]. Furthermore, based on academic studies commercial analytical instruments have been developed for the sensitive quantitation of proteins[18,19] and for DNA sequencing[20,21]. However, the techniques are mainly developed for laboratory use, with samples being processed through multiple steps and biochemical reagents, which is not suited for continuous in-situ monitoring applications.

In this paper, we demonstrate biomarker monitoring based on the sensing of particle mobility (BPM). BPM is a sensing strategy for continuous monitoring that is (i) based on single-molecule resolution, (ii) self-contained and stable because it does not consume or produce any reactants, and (iii) based on affinity interactions and thereby suited for a diversity of biomarkers. BPM is based on particles that are coupled to a substrate via a flexible molecular tether, with both the particles and substrate provided with affinity molecules for effectuating specific and reversible interactions. Digital state-change events of the particles are observed and lifetimes and switching activity are analyzed. The particles dynamically respond to increases and decreases of target concentration and the relaxation times are related to the kinetic characteristics of the affinity molecules. The technology is demonstrated with DNA and protein as model biomarkers, in buffer and in blood plasma, showing sensitivity to picomolar and nanomolar concentrations. The selectivity of BPM is demonstrated by measuring in complex media and by comparing hybridization with single basepair differences. With its basis in reversible affinity interactions and single-molecule resolution, we envisage that the presented technology will pave the way for continuous monitoring of a wide variety of biomarkers with high sensitivity, specificity, and accuracy.

## Results

**Design of biomarker monitoring based on particle mobility**. The design relies on measuring particle mobility modulated by single-molecule interactions (see Fig. 1). The particles and substrate are kept close to each other by a flexible dsDNA tether, and they are both provided with binder molecules that have specific and reversible affinity interactions with the target molecules. We refer to binders coupled to the particle as capture molecules (selected to have a high affinity) and to binders on the substrate as detection molecules (having lower affinity to the target). When a target becomes bound to a capture as well as to a detection molecule, a compact molecular sandwich is formed, leading to a bound state of the particle. The bound state has a reduced particle mobility that is recorded using dark-field optical video microscopy (see Fig. 1a, b). The Brownian particle motion is easily observed due to the very large and stable optical scattering signals of the particles, which makes the detection practical and reliable. The reduction of particle mobility is transient in nature due to the reversibility of the single-molecule affinity interaction, generating a digital switching signal (see Fig. 1c). At a low target concentration, the particle is mostly unbound and rarely bound. When more targets are present, the chance that the particle binds to the substrate increases and therefore also the frequency of switching between bound and unbound states. Therefore, the switching activity of the system, i.e. the number of binding and unbinding events per particle per time interval, is a measure for the concentration of target molecules in solution. The activity is determined as:

$$\text{Activity} = \frac{N_{\text{events}}}{N_{\text{particles}} \cdot T_{\text{measurement}}}, \qquad (1)$$

with $N_{\text{events}}$ as the number of observed events, $N_{\text{particles}}$ the number of particles, and $T_{\text{measurement}}$ the duration of the measurement.

The experiments involve a specific nucleic acid and the protein thrombin as target molecules. Circulating nucleic acids such as DNA, mRNA, and microRNA (miRNA) are an upcoming class of biomarkers for non-invasive diagnostics as their presence or change of levels could indicate disease progression[22,23]. Our study involves detection of the DNA equivalent of miRNA-126 through DNA hybridization. miRNA-126 is a short non-coding RNA molecule which can function as both tumor suppressor and oncogene depending on the types of cancer[24,25], and their circulation in the blood stream is ideal for non-invasive diagnostics of disease progression[22]. We also demonstrate the detection of thrombin using a pair of DNA aptamers. The thrombin aptamers are short DNA (15-mer and 29-mer) oligonucleotides that bind to different epitopes on the thrombin protein, such that a sandwich assay format can be used. Thrombin is a key protease in the blood coagulation cascade and is present in blood in its inactive prothrombin form at normal conditions[26,27]. Both thrombin and prothrombin are involved in many pathological conditions, such as Alzheimer's disease and cancer[28].

**Analysis of particle mobility time-traces and state lifetimes**. To prove that the experiment of Fig. 1 is able to resolve discrete binding events, we recorded mobility traces and analyzed state lifetimes for both the DNA and the thrombin system. Figure 2a shows an example of a recorded mobility trace of a particle, where

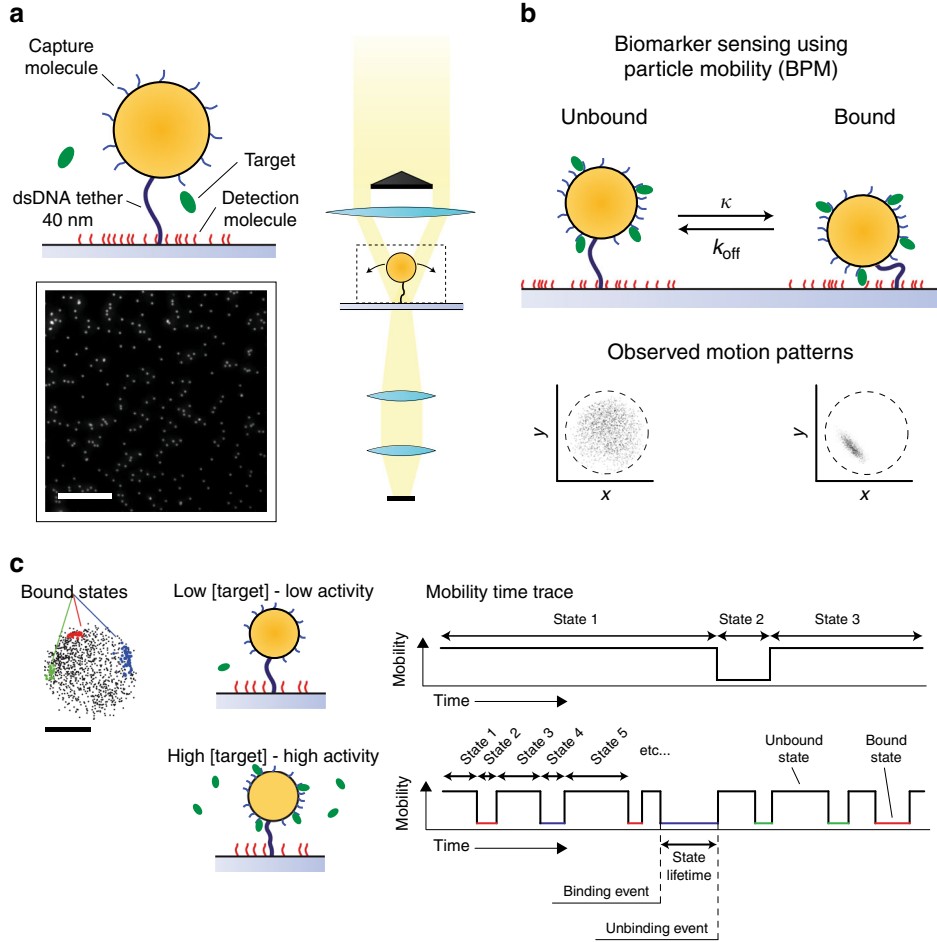

**Fig. 1** Biomarker monitoring based on the sensing of particle mobility (BPM). **a** Particles (orange) are tethered to the substrate via a 40 nm dsDNA strand (black). The particles are functionalized with capture molecules (blue), which serve to capture the target molecules from solution. The substrate is functionalized with lower-affinity detection molecules (red), which create short-lived target-induced bonds between the particle and the substrate. Target molecules (green) are either 22-nucleotide ssDNA molecules or thrombin proteins. In the experiments, the high-affinity capture molecules were biotinylated and coupled to the particles by biotin–streptavidin coupling; the low-affinity detection molecules were provided with a biotin tag and coupled to the substrate by neutravidin or streptavidin (see Methods). The particles are detected using darkfield microscopy, imaging the particles as bright dots on a dark background. The scale bar represents 50 μm. **b** Target binding causes the particle to become intermittently bound to the substrate resulting in switching between different mobilities and motion patterns. The effective association rate of the particle to the substrate is indicated as $\kappa$[30], the dissociation rate is given by $k_{off}$. **c** The mobility of the particles is analyzed as a function of time and the binding/unbinding events are digitally detected for hundreds of particles in parallel. The time between two consecutive events corresponds to the lifetime of the enclosed state. Two example mobility traces are sketched for a particle with no target molecules in solution or with a high target molecule concentration, leading to a low or a high observed switching activity. The scale bar represents 250 nm

the optically observed in-plane position $(x, y)$ and the step size $(\Delta x\, \Delta y)$ were recorded as a function of time. The motion pattern of the particle shows a disc-like shape (black dots) with four smaller dense areas on the periphery (orange, blue, purple, black). The disc-like pattern indicates that this particle is attached to the substrate by a single dsDNA tether and is mostly unbound[29]. The four smaller areas relate to short-lived bound states with confined motion. The switching behavior can be understood by statistically analyzing the state lifetimes. We developed an algorithm to automatically detect state switch events in the measured time traces (see Methods section). The algorithm finds the times at which the mobility switches by scanning the data for statistically significant deviations. Each vertical red line in Fig. 2a marks a switching event, i.e. a significant change in the mobility of the particle. From the switching events, the state lifetimes can be determined, corresponding to the timespan between two consecutively detected switching events, as sketched in Fig. 1c. From the recorded lifetimes, cumulative lifetime distribution functions

or survival curves were constructed, for both the DNA and the thrombin systems (see Fig. 2b).

The survival curves contain both bound state lifetimes, as well as unbound state lifetimes. The observed motion states could not be directly assigned as being bound or unbound states. For example, short unbound states can look similar to bound states in terms of lifetime and particle motion parameters, as bound states have variable appearances due to particle surface roughness and different locations of capture and detection molecules. Therefore, the survival curves were fitted with a double-exponential function so that both characteristic lifetimes could be extracted from a single fit.

The fits are double-exponential curves of the form:

$$a \cdot \exp(-k_{off} \cdot t) + b \cdot \exp(-\kappa \cdot t). \qquad (2)$$

The bound and unbound state lifetimes are then

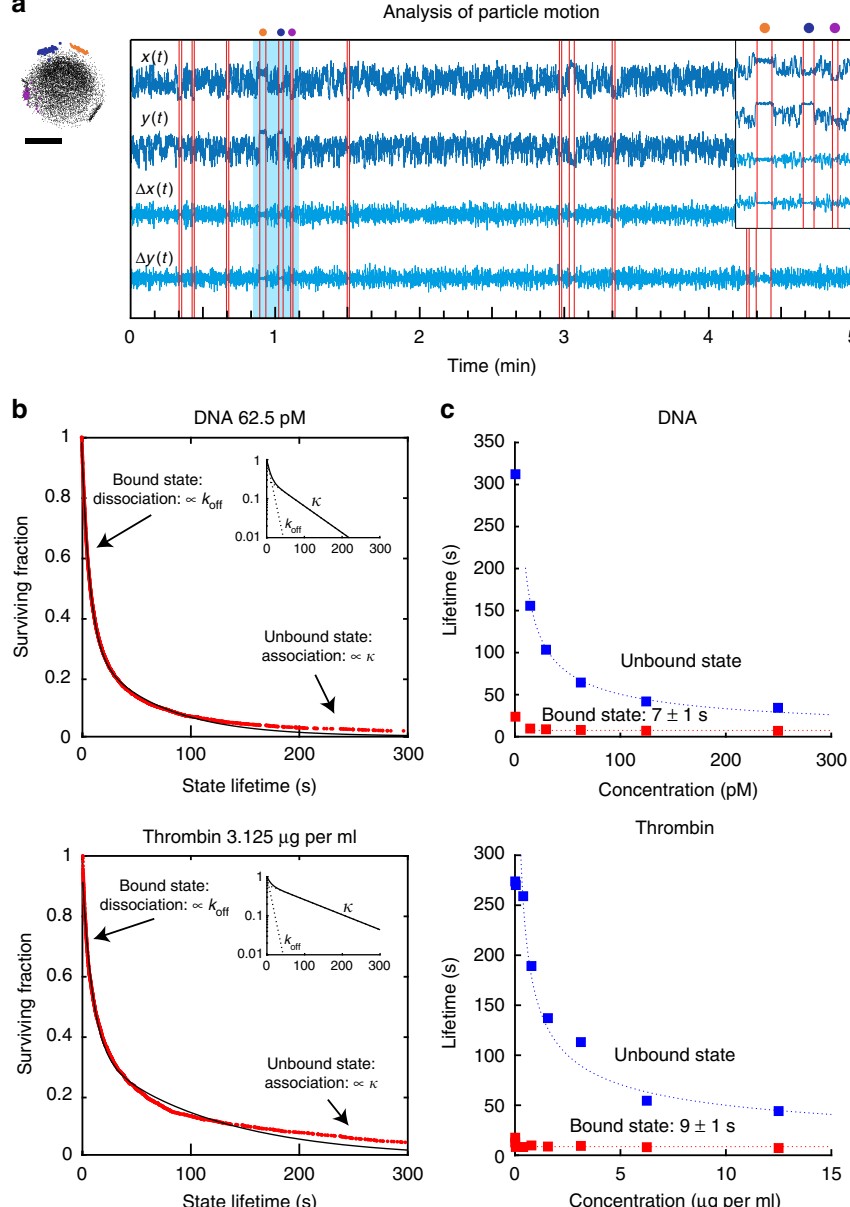

**Fig. 2** Analysis of mobility time-traces in terms of state lifetimes. **a** The mobility of a single particle with switching activity. The optically recorded in-plane position $(x, y)$ and the step size $(\Delta x, \Delta y)$ are shown as a function of time. The detected times at which the motion is observed to change are indicated with vertical red lines. The timespan between two consecutive change events corresponds to a single-state lifetime of the particle. The motion pattern is shown of the particle that was analyzed. An expanded view of time traces with three bound states is shown in the inset, corresponding to the timespan marked by the light blue area in the main graph. The confined motion patterns of three bound states are indicated with colored data points. In the time trace three corresponding bound states are indicated with dots above the graph. The scale bar represents 250 nm. **b** State lifetime analysis for ssDNA and thrombin experiments, graphed as survival curves. States with a lifetime of 1 s or more have been analyzed. The lifetimes were determined from the analysis of the motion data during a 5-min measurement at a target concentration of 62.5 pM DNA (501 tracked particles) or 3.125 μg/mL thrombin (407 tracked particles), combining both unbound states and bound states in a single survival curve. Red curves represent data, black lines represent fits. The insets show the fit curves and the individual contributions of the time constants $k_{off}$ and $\kappa$; the lin-log scale highlights the double-exponential character of the decay. For DNA: $\tau_{bound} = 7.3 \pm 0.1$ s and $\tau_{unbound} = 63.8 \pm 0.4$ s; for thrombin: $\tau_{bound} = 9.3 \pm 0.1$ s and $\tau_{unbound} = 113 \pm 2$ s. **c** Dependence of lifetimes of the bound (red) and unbound (blue) states on the target concentration. The error related to the stochastics of the measurement is smaller than the symbol size; experimental contributions to the measurement errors are discussed in Supplementary Note 1. The dotted lines indicate the approximate trend of the characteristic lifetimes. The dotted blue lines scale with the target concentration as $\sim [T]^{-0.5}$ for DNA and thrombin. The dotted red lines do not depend on the concentration (see also Supplementary Notes 2 and 3)

determined using:

$$\tau_{\text{bound}} = \frac{1}{k_{\text{off}}}, \qquad (3)$$

$$\tau_{\text{unbound}} = \frac{1}{\kappa}. \qquad (4)$$

The fits (black lines) overlap largely with the data (red curves), indicating that the switching behavior results from interactions with two distinct kinetic rate constants, namely slow association ($\kappa$) and fast dissociation ($k_{\text{off}}$)[30,31]. The fitted lifetimes are shown as a function of the target concentration in Fig. 2c. The data shows that the unbound state lifetimes, which correspond to an association rate, clearly depend on the target concentration. For low target concentrations, the unbound state lifetimes are

determined by the non-specific binding rate between particle and substrate (lifetime of about 300 s) and for increasing target concentration the unbound state lifetimes decrease. This is consistent with the interpretation that more target molecules are captured on the particle and are therefore available for the formation of sandwich bonds between particle and substrate.

**Dose–response curves and sensor reversibility**. We now apply to the data the concept of particle-switching activity, which is the number of observed state-switching events per particle during the measurement (see Eq. (1)). The activity is a counting statistic that can be readily determined for all particles, analogous to the rate of counts in measuring radioactivity. The switching activity has been studied as a function of target concentration and time, for monitoring of the 22-nucleotide ssDNA and the thrombin protein target (see Fig. 3). The reported activities are averages

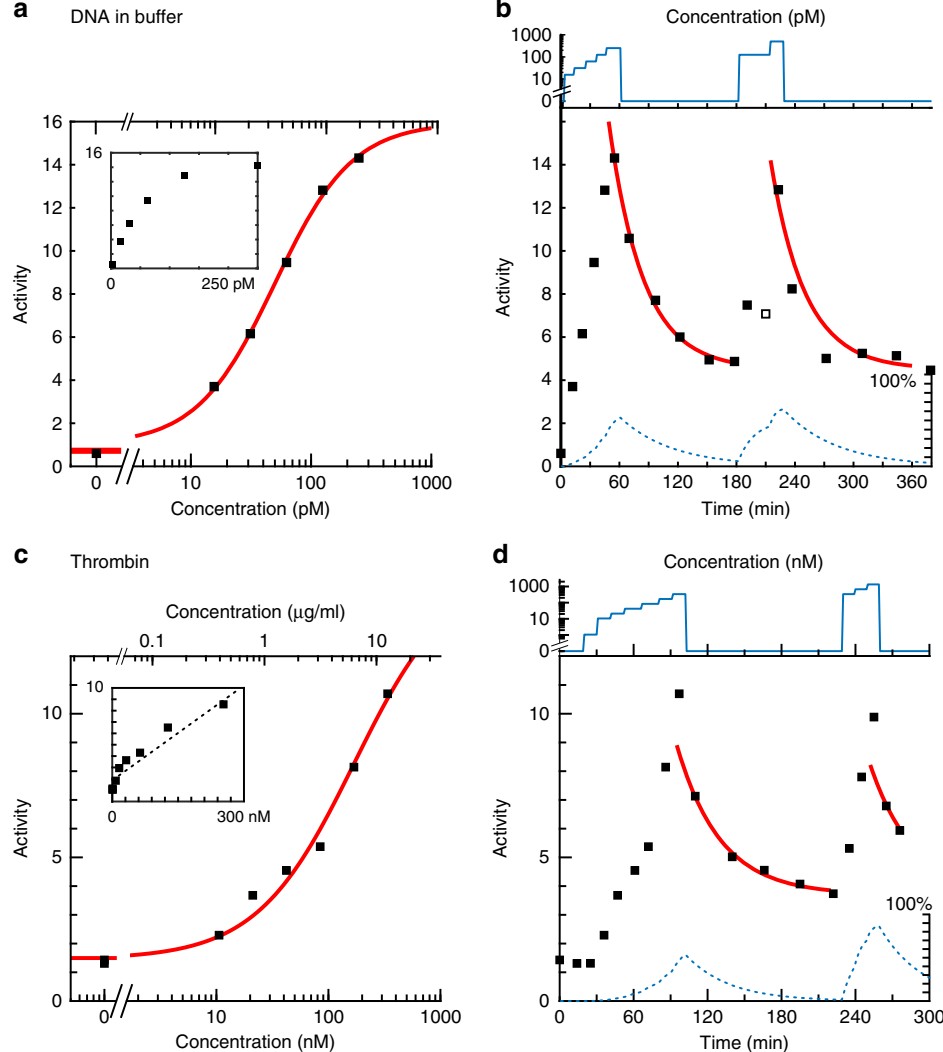

**Fig. 3** Particle-switching activity as a function of target concentration and time, for monitoring ssDNA and thrombin protein, for the assays of Fig. 2. **a**, **c** Average number of detected motion changes per particle per 5 min as a function of the target concentration. The error related to the stochastics of the measurement is smaller than the symbol size; experimental contributions to the measurement errors are discussed in the Supplementary Note 1. The data is fitted by the Hill equation with a baseline offset (red line). The insets show the dose–response curves on double-linear axes. **b**, **d** Sensor reversibility is demonstrated by recording the particle activity as a function of time. The graph shows the applied target molecule concentration and the measured particle activity as a function of time. In the first section (up to 60 min in, up to 100 min in) the target molecule concentration was increased in a stepwise fashion. Thereafter the concentration was kept at zero for 120 min and subsequently pulsed from zero to a high concentration. One measurement point (open square) is considered to be an outlier as it deviates from all measurements before and after. Fits (red solid lines) represent exponential decay of the activity data using the model described by Eq. (6). The background activity is the same for both fits in the measurement series. The dotted curves illustrate the activity (on a scale from 0 to 100%) predicted by a reaction kinetics model that is described in the Supplementary Note 4

over about 400 particles that are measured simultaneously, with a measurement duration of 5 min. Particles with strong irregularities were excluded from the analysis, e.g. particles with strongly confined or asymmetrical motion (see Methods section).

Panels a and c of Fig. 3 show curves of activity versus concentration, and panels b and d the complete time traces including reversibility. For both molecular systems, the activity at zero target concentration is non-zero, caused by temporary non-specific interactions between particle and substrate[32]. The activity versus concentration curves show a monotonic increase in the activity of the particles with increasing concentrations of the target molecules. The curves exhibit an S-shape that relates to the binding of target molecules to the capture molecules on the particles. The curves have been fitted with a function based on the Hill equation that describes the equilibrium behavior of reversible biomolecular binding:[31]

$$A = A_{\mathrm{b}} + A_{\mathrm{ampl}} \frac{[T]^n}{K_{\mathrm{d}}^n + [T]^n}, \qquad (5)$$

where $A$ is the switching activity, $[T]$ is the target concentration, $K_{\mathrm{d}}$ the apparent equilibrium dissociation constant of the interaction between capture molecule and target, $n$ is the slope factor, $A_{\mathrm{ampl}}$ is the activity amplitude, and $A_{\mathrm{b}}$ is the baseline activity. The fitted dissociation constant of the DNA curve is $K_{\mathrm{d,DNA}} = 49 \pm 10$ pM. The thrombin data was measured for bovine thrombin (Fig. 3) as well as for human thrombin (see Supplementary Figure 1). The fitted dissociation constants are $K_{\mathrm{d,thrombin\ (bovine)}} = 170 \pm 56$ nM and $K_{\mathrm{d,thrombin\ (human)}} = 250 \pm 60$ pM; the latter corresponds well to the reported value in literature for the 29-mer capture aptamer[33]. Both the DNA and the thrombin data is fitted with $n = 1$.

A continuous monitoring system needs to generate a signal that follows the target concentration over time. The dynamic monitoring capabilities of the BPM technology are studied in Fig. 3b, d. The applied concentrations of DNA and thrombin are indicated as a function of time (note the logarithmic concentration scales). After exposure to a series of stepwise increases of target concentration, a flush with target-free buffer was applied. Thereafter the activity of the system was measured over an extended period of time in order to record the reversibility behavior, corresponding to the release of target molecules from the capture molecules. The decrease in the activity of the system was modeled as a function of time $t$ using an exponential decay function:

$$A_{\mathrm{b}} + b \cdot \exp\left(-\frac{t}{\tau_{\mathrm{off}}}\right), \qquad (6)$$

with a time-constant $\tau_{\mathrm{off}}$, an amplitude of $b$ and a background activity $A_{\mathrm{b}}$.

The DNA system returns closely to the activity level at $t = 0$, indicating that all target molecules have been released from the particle. The thrombin system stabilizes at a somewhat higher activity level, which may indicate that not all thrombin molecules were released from the particles due to multivalent or non-specific interactions. Both measurement series end with another concentration pulse followed by a second flush with target-free buffer, demonstrating once again the reversibility of the sensing systems. The second peaks have a somewhat lower activity than the first ones, which may indicate that the probes in the system become less effective over time; current ongoing work is investigating this issue. The red lines represent single-exponential relaxation curves, with $\tau_{\mathrm{off}} = 35 \pm 5$ min for the DNA system and $\tau_{\mathrm{off}} = 35 \pm 15$ min for the thrombin system. The relaxation is the result of target molecules unbinding from

capture molecules and subsequently diffusing out of the sensing region between particle and substrate. Due to the confinement between particle and substrate, locally the effective concentration of molecules is high and may be on the order of μM to mM, so well above the $K_{\mathrm{d}}$. This can cause a nonzero target rebinding probability that slows down the diffusive escape process[34]. As the measured relaxation times are larger than bimolecular dissociation times found in literature, it is indeed possible that rebinding plays a role in these experiments.

The dashed blue lines in Fig. 3b, d show activity curves modeled using simple rate equations of the interaction between the target and capture molecules (see Supplementary Note 4). The activity is calculated from the known target concentration, the values of $K_{\mathrm{d}}$ determined in Fig. 3a, c, and the values of $k_{\mathrm{off}}$ determined in Fig. 3b, d. In spite of its simplicity, the model qualitatively reproduces the measured increases and decreases of activity with target concentration. The calculated activities increase monotonically during the stepwise increase of the target molecule concentration, indicating that the assays occur in a kinetic regime rather than in full thermodynamic equilibrium. The theoretical relaxation time of the system scales in first approximation as:

$$\tau = \frac{1}{k_{\mathrm{off}} + k_{\mathrm{on}}[T]}, \qquad (7)$$

so it decreases with increasing target concentration (see Supplementary Note 4). It also shows that the response time of the system has the potential to be tuned through the affinity of the binder molecule. These results show that the BPM technique has a dynamic response to increases as well as decreases of the concentration of target molecules, demonstrating its potential for applications wherein biomarker concentrations need to be monitored over time.

**Continuous monitoring of ssDNA in filtered blood plasma.** Next, we demonstrate the capabilities for measuring in biologically relevant media, by sensing target ssDNA in undiluted blood plasma. Since continuous monitoring applications typically employ porous membranes[7,35], a 50 kDa cut-off filter was used which lets sub-50-kDa biomolecules pass but blocks larger proteins and protein aggregates. The results are presented in Fig. 4. The outcomes of the experiment are striking in three ways. First, the background activity level before addition of target is equally low in buffer and in undiluted filtered blood plasma ($A \sim 0.6$ at 0 pM in Figs. 3b and 4b) showing that non-specific interactions are well-suppressed by the developed surface chemistries of particle and substrate. An experiment using blood plasma without any filtration showed higher non-specific binding, so operation in unfiltered plasma will require optimizations of antifouling properties. Second, the data measured in buffer (Fig. 3a, b) and in plasma (Fig. 4b, c) are very similar, in terms of the dose–response curve as well as in reversibility and relaxation time, showing that the affinity molecules function properly in both matrices. The dose-response curve in plasma is shifted to a slightly higher concentration, indicating that there might be a weak blocking effect from the plasma. Third, the data show sensitivity to pico-molar target concentrations, which is impressive for a measurement technique that does not employ any reagents nor any biochemical amplification.

**Selectivity of BPM studied with ssDNA.** The experiments in blood plasma demonstrate the selectivity of biomarker detection by BPM. The high selectivity can be understood by the double binding of the sandwich assay configuration, where signals result from the simultaneous occurrence of two molecular bonds:

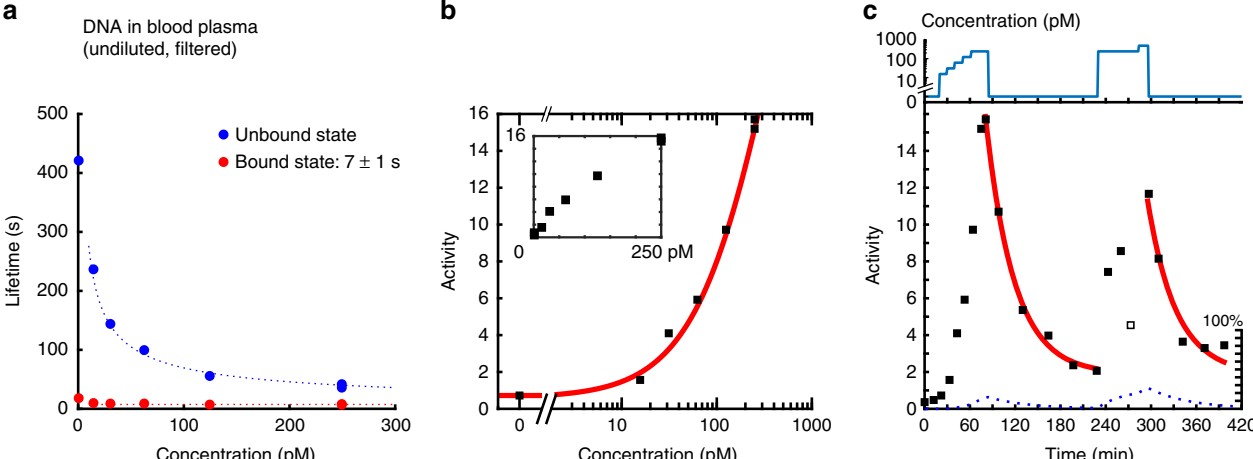

**Fig. 4** Monitoring of ssDNA in undiluted filtered blood plasma. **a** Lifetimes of the bound and unbound states as a function of the target concentration. The dotted blue line scales with the target concentration as $[T]^{-0.6}$. The dotted red line does not depend on the concentration. **b** Activity as a function of concentration, recorded during the stepwise increase in concentration. The fit does not yield a reliable $K_d$ value due to the absence of saturation in the curve. The inset shows the dose–response curve on double-linear axes. **c** Sensor reversibility is demonstrated by recording the particle activity as a function of time. The graph shows the target molecule concentration, as well as the measured particle activity as a function of time. In the first section (up to 80 min) the target molecule concentration was increased in a stepwise fashion. Thereafter the concentration was kept at zero for 150 min and subsequently pulsed from zero to a high concentration and brought back to zero again. The data point with open square is considered to be an outlier, as it deviates from measurements before and after. Fits (red solid lines) represent exponential decay curves of the activity using Eq. (6), with a time-constant $\tau_{off}$ and background $A_b$ that is the same for both fits in the measurement series. A time constant $\tau_{off} = 37 \pm 6$ min was found in blood plasma. The dotted curves illustrate the activity (on a scale from 0 to 100%) predicted by the reaction kinetics model (see Supplementary Note 4)

between target molecule and capture molecule (on the particle) and between target molecule and detection molecule (on the substrate). The selectivity was also apparent in the thrombin experiment of Fig. 3c, d, as that experiment was performed with a solution containing a high concentration of background protein (the BSA concentration in solution was 20,000 times higher than the lowest thrombin concentration). The selectivity is further demonstrated by the DNA experiments in Fig. 5. Fig. 5a shows three different detection molecules that were used in the BPM system and were exposed to DmiRNA-126 target, i.e. the DNA equivalent of miRNA-126. Perfect match is indicated in blue while mismatch is indicated in red. Figure 5b shows the activity measured before and after injection of 100 pM DmiRNA-126. The data show that an activity increase is only observed for 9 bpD, the detection molecule with a perfect 9 bp match, and not for 8 bp perfect match (8 bpD) nor for 9 bp with one mismatch (9 bpMM). This demonstrates that the BPM system is sensitive to single-base pair hybridization differences. Figure 5c shows an experiment in which the BPM system was sequentially exposed to two different targets, namely a random DNA sequence having no complementarity with either capture molecule or detection molecule, and thereafter the 100 pM DmiRNA-126 target. Again, an order of magnitude activity change is observed for the specific target, and no activity increase is seen for the random sequence.

These experimental results convincingly demonstrate that BPM allows biomarker monitoring even at low picomolar concentrations, that it is compatible with complex biological media, and that the measurement technique is highly selective, which represents a unique set of properties for the field of continuous biomarker monitoring.

## Discussion

A very important property of BPM is that affinity interactions are used. The benefit of using affinity binders such as aptamers, antibodies, recombinant antibody fragments[36,37], or non-antibody scaffolds[38], is that a broad variety of biomarkers can be addressed and that the sensitivity range and response time can

be tuned by the $K_d$ of the capture molecules. Furthermore, the affinity-based sensing principle is direct and self-contained, without consuming or producing any reactants, which has benefits for biocompatibility and long-term stability.

Another important property of BPM is that the readout technology is digital, based on discrete changes of particle mobility, resulting from single-molecule binding and unbinding events. Having clear digital signals provides benefits for sensor design and operation, reminiscent of the advantages that have fueled the transition from analog to digital in fields such as electronics. Beneficial properties are reproducibility, internal calibration, accuracy, and selectivity. To briefly mention the aspect of selectivity: The signals of traditional biosensors are the sum of many specific and non-specific molecular events, recorded over large areas and long timescales; this summation masks the underlying molecular-scale dynamic processes. In contrast, single-molecule methods reveal time-dependent signals from individual molecules that are generally different for specific and non-specific interactions. Therefore single-molecule analysis in principle allows a separation of specific and non-specific signals by kinetic fingerprinting[17], thereby enhancing biosensing selectivity.

It is interesting to compare BPM to other single-molecule-based biosensing assays that could possibly be used for in-situ monitoring applications. Compared to single-molecule fluorescence[17], BPM uses particles that give optical signals which are orders of magnitude larger than signals from single fluorophores, in terms of optical cross-section as well as signal-to-noise ratio of single-molecule (un)binding events. Furthermore, BPM is not hampered by blinking or bleaching of fluorophores, and BPM is fully surface-bound without soluble probes, in contrast to PAINT[14]. Compared to carbon nanotube luminescence[39], BPM has optical signals that are orders of magnitude larger than those produced by carbon nanotubes. The large and stable optical signals of BPM allow extraction of reversible bond lifetimes and switching statistics, which is essential for a monitoring biosensor with single-molecule resolution. In addition, the described carbon nanotube sensing relies on aptamer folding[39], which limits the

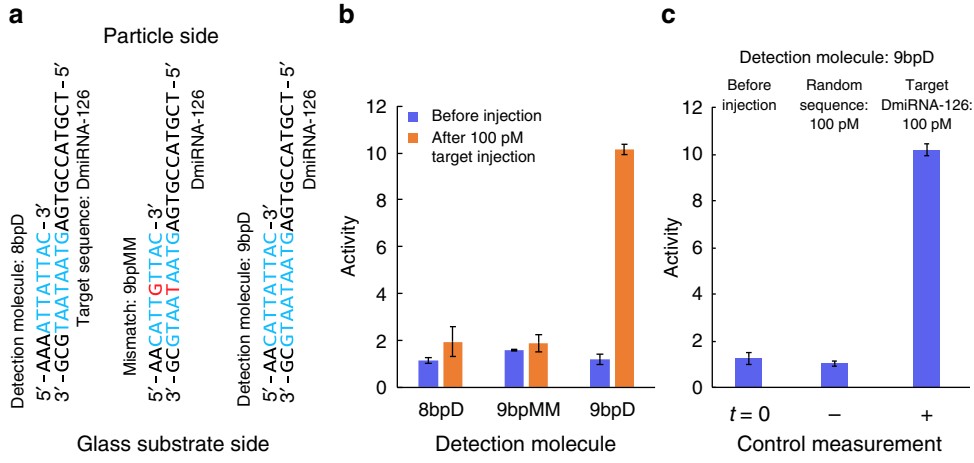

**Fig. 5** Selectivity of detection of the DNA equivalent of miRNA-126, i.e. DmiRNA-126. **a** The target molecule with three different detection molecules. 8 bpD is a detection molecule with 8-bp perfect match to the target. 9 bpMM is a detection molecule with 8-bp perfect match and a single mismatch in the middle. 9 bpD is a detection molecule with 9-bp perfect match with the DmiRNA-126 target. Perfect match is indicated in blue and mismatch is indicated in red. **b** Detection of 100 pM DmiRNA-126 with 8 bpD, 9 bpMM, and 9 bpD detection molecules. The used capture molecule on the particle has 11-bp complementarity to the target. Particle switching activities were recorded before and after supply of 100 pM DmiRNA-126 target. **c** Particle switching activity using 9 bpD detection molecule and different target oligonucleotide sequences. Sequentially, particle-switching activities were recorded when there was no target injected, when 100 pM target with a random DNA sequence was injected, and finally when 100 pM DmiRNA-126 target was injected. Error bars are standard deviations of two independent measurements with 5 min duration

applicable set of biomarkers. BPM is based on just affinity binding, so that a broad variety of biomarkers can be addressed using the vast range of existing affinity molecules. Nanopore single-molecule biosensing relies on detecting ionic currents or charge[40] and therefore suffers interference from ions, charged molecules, and redox molecules; this makes detection in biological media far from trivial. Furthermore, for acquiring statistics, scaling of BPM (imaging a substrate with many particles) is simple and cost-effective compared to nanopore scaling (arrayed multilayer chip). Compared to refractive index methods, such as nanoplasmonics[41] and interferometric scattering (iSCAT)[42], BPM has optical signals that are orders of magnitude larger. Furthermore, refractive index methods accumulate background signal from all macromolecules in solution, while BPM does not. Finally, single-molecule tweezers assays have affinity binders only in the tether[43], which gives low target capture rates per particle. BPM has a higher number of affinity molecules per particle, resulting in higher target capture rates. Furthermore, BPM is more scalable and simpler because it does not require force application, nor distance detection along a force coordinate. Therefore, taking all these considerations into account, we conclude that BPM has unique properties and clear advantages for continuous biomarker monitoring.

In this work we have studied DNA as a model system because the molecules can be ordered commercially and affinities can be engineered by hybridization complementarity. The sensitivity of BPM demonstrated in blood plasma is very high (picomolar) but not yet high enough to detect circulating miRNAs in clinical samples. Before that, we think that the BPM technology will become clinically relevant for detecting hormones, drugs, peptides, and proteins, because many of these biomarkers are present in the nanomolar and picomolar concentration ranges.

The results presented in this paper open a wide range of possibilities for further research and applications. Important next steps will be to further expand the range of capture molecules (e.g. antibodies, nanobodies), assay formats (e.g. competitive), and matrices (*e.g.* interstitial skin fluid). The individual particles yield sizable optical signals, so that the readout system can be strongly miniaturized, to a degree that it will become suited for on-body and even in-body applications. With its basis in affinity

binding and single-molecule resolution, we envisage that the presented technology will enable biosensors for continuous biomarker monitoring with high sensitivity, specificity, and accuracy.

## Methods

**Particle and surface functionalization for the DNA system.** Ten microliters of the streptavidin-coated magnetic particles (10 mg/mL, Dynabeads MyOne Streptavidin C1, 65001, Thermo Scientific) was incubated with 10 μL 120 bp dsDNA tether (with 5′ Digoxigenin and 5′ Biotin on either end) at a concentration of 2 nM in PBS buffer solution for 10 min on rotating fins (VWR, The Netherlands). Next, 10 μL PBS with 10 μM of capture molecule (11-nt oligo, 5′-TCACGGTACGA-3′ Biotin, Integrated DNA Technologies) was added to the particle mixture and incubated on rotating fins for 30 min. Then 10 μL mPEG-Biotin (PG1-BN-1k, Nanocs) of 100 μM in PBS were incubated with the particle mixture for 5 min. After that, the particle mixture was washed three times with 500 μL PBS, reconstituted in 630 μL PBS/BSA buffer (PBS with 1% BSA filtered with a 0.22 μm filter and degassed in a vacuum desiccator) and kept on the rotating fin for 30 min. Finally, the particle mixture was sonicated with 10 pulses at 70% with 0.5 duty cycle (Hielscher, Ultrasound Technology).

Functionalization of the glass substrate was performed with the measurement chamber integrated into the flow setup. Details on the flow system are given in Supplementary Figure 2 and Supplementary Note 5. After PBS wetting of the flow cell, 350 μL PBS with 100 μg/mL neutravidin (31000, Thermo Scientific) and 50 ng/mL anti-Digoxigenin antibody (ab76907, Abcam) was aspirated into the system and incubated for 1 h. Then 350 μL PBS with 500 nM detection oligo (9bpD: 5′ Biotin-AACATTATTAC-3′, 8bpD: 5′ Biotin-AAAATTATTAC-3′, and 9bpMM: 5′ Biotin-AACATTGTTAC-3′, Integrated DNA Technologies) was aspirated into the system and incubated for 30 min. Then 350 μL PBS/BSA with 20 μM Biotin–PEG was aspirated into the system and incubated for 15 min. After that, 350 μL PBS/BSA buffer was aspirated into the system and finally 350 μL particle mixture prepared before were aspirated into the system and incubated for 1 h. After that the system was ready for measurement. The flow speed was 100 μL per min.

After assembly of the tethered particles on glass slides, different concentrations of target molecules (DmiRNA-126: DNA equivalent of miRNA-126, 5′-TCGTACCGTGAGTAATAATGCG-3′, Random DNA sequence: 5′-TTTGTTAGACGGACACTGTATGATTT-3′, Integrated DNA Technologies) in PBS were injected into the flow cell. For each concentration, 400 μL of the solution was injected into the measurement chamber at a rate of 100 μL per min, after which the particle motion was recorded for 5 min in the absence of flow. The starting time of each measurement was recorded.

**Particle and surface functionalization for the thrombin system.** Ten μL of the streptavidin-coated magnetic particles (10 mg/mL, Dynabeads MyOne Streptavidin C1, 65001, Thermo Scientific) was incubated with 10 μL 120 bp dsDNA tether (with 5′ Digoxigenin and 5′ Biotin on either end) at a concentration of 2 nM in PBS buffer solution for 10 min on rotating fins (VWR, The Netherlands). Next, 10 μL PBS with 50 nM of the 29-mer aptamer (5′ Biotin-TTTTTTTTTTTTTTTTAGTC

CGTGGTAGGGCAGGTTGGGGTGACT-3′, Integrated DNA Technology) was added to the particle mixture and incubated on rotating fins for 30 min. Then 10 μL mPEG–Biotin (PG1-BN-1k, Nanocs) of 100 μM in PBS were incubated with the particle mixture for 5 min. After that, the particle mixture was washed three times with 500 μL PBS and reconstituted in 630 μL assay buffer (PBS with 1% BSA and 0.05% Tween 20 (Sigma Aldrich), filtered with a 0.22 μm filter and degassed in the desiccator) and incubated on the rotating fin for 30 min. Finally, the particle mixture was sonicated with 10 pulses at 70% with 0.5 duty cycle to break up particle clusters that may have formed.

The glass substrate was prepared in the flow cell as in the DNA experiment with minor adjustment. After PBS wetting of the flow cell, 350 μL PBS with 1 μM streptavidin (S4762-1MG, Sigma Aldrich) and 50 ng/mL anti-digoxigenin antibody (ab76907, Abcam) was aspirated into the system and incubated for 1 h. Then 350 μL PBS with 500 nM 15-mer detection aptamer (5′ Biotin-TTTTTTTTTTTTT TTTGGTTGGTGTGGTTGG-3′, Integrated DNA Technology) was aspirated into the system and incubated for 30 min. Then 350 μL PBS/BSA with 20 μM Biotin–PEG was aspirated into the system and incubated for 15 min. After that, 350 μL assay buffer was aspirated into the system and finally 350 μL particle mixture prepared before were aspirated into the system and incubated for 1 h. After that the system was ready for measurement.

Thrombin protein (T4648-1KU, Sigma Aldrich) was diluted in assay buffer at different concentrations and aspirated into the system with a volume of 500 μL from low concentration to high concentration or according to the time series described in the Results section. The flow speed in the thrombin experiment was 100 μL per min.

**DNA detection in blood plasma**. Bovine plasma was purchased from Sigma Aldrich (P4639-10ML) and reconstituted in 10 mL MilliQ water. The plasma was filtered through a 50-kDa molecular cut-off centrifugal filter (UFC905008, Millipore). The plasma filtrate was collected and spiked with the DNA equivalent of miRNA-126 (5′-TCGTACCGTGAGTAATAATGCG-3′, Integrated DNA Technologies) at the concentration indicated in the results. The measurements were then performed as in the other experiments.

**Detection and analysis of particle motion**. Particle motion was recorded on a Nikon Ti-E inverted microscope (Nikon Instruments Europe BV, The Netherlands), at a total magnification of 20× using an iXon Ultra 897 EMCCD camera (Andor, Belfast, UK). The particle motion in a field of view of $405 \times 405\ \mu m^2$ was recorded for 5 min at a frame rate of 30 Hz under dark-field illumination conditions. Tracking of the particles was performed by determining the center-of-intensity of the bright particles on the dark background.

The shape of the motion pattern of a tethered particle was quantified in terms of the motion amplitude and the motion symmetry. The amplitudes of the major ($A_{major}$) and minor ($A_{minor}$) axis of motion were determined by calculating the covariance matrix of the position data[44]. From the amplitudes the symmetry parameter

$$S_{sym} = A_{minor}/A_{major} \qquad (8)$$

was calculated. Particles with strong irregularities were excluded from the data analysis, namely particles with too large motion ($A_{minor} > 150$ nm) or with asymmetrical motion ($S_{sym} < 0.33$). Particles with a confined ring-shaped motion pattern[32] were excluded by defining a radial confinement parameter

$$Rc = \sigma_r/\bar{r}, \qquad (9)$$

where $r$ is the radial position of the particle and exclusion condition Rc < 0.2.

Trajectory analysis was performed on all individual particles in order to detect binding and unbinding events. An algorithm was developed in Matlab to automatically scan through the motion data and detect changes in the distribution of the particle positions. The algorithm uses the second moment divergence function $C(n)$ to detect changes in particle motion and determine the time-point of state change. Deviations are identified via

$$C(n) = \sum_{i=1}^{n}(x_i - \mu)^2 - \sigma^2, \qquad (10)$$

where $x_i$ are the data points, $\mu$ is the average value and $\sigma$ is the standard deviation of the data in a reference interval, and $i$ enumerates the data points following the reference interval. $C(n)$ is a significance measure for an observed deviation. Back-extrapolation of deviations in $C(n)$ to zero is used to identify the time-point of change. The analysis was performed for the position data of the particles, in Cartesian as well as polar coordinates. The algorithm was tested on experimental data and simulated data, showing a time-point accuracy below 0.1 s. Events with a state duration over 1 s are reliably detected. Further details on the algorithm will be described in a separate publication.

**Data availability**. The data that support the findings of this study are available from the corresponding author upon reasonable request.

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

## Acknowledgements

We thank Peter Zijlstra, Maarten Merkx, and Lorenzo Albertazzi for critically reading the manuscript. J.Y. acknowledges support from European Union Marie Skłodowska-Curie Action Individual Fellowship number 703523.

## Author contributions

All authors conceived the measurement system and experiments. E.V. and J.Y. are equally contributing first authors. E.V. and J.Y. performed the experiments and the analysis of the data. E.V. conceived the algorithm for the analysis of the data. All authors co-wrote the paper.

## Additional information

**Competing interests:** M.P., L.J.v.I., and E.V. are listed as inventors on patent application WO/2016/096901 "Biosensor based on a tethered particle". The remaining authors declare no competing interests.

