## [Peer Review File · Nature Communications]

REVIEWERS' COMMENTS:

Reviewer #1 (Remarks to the Author):

The comments from the reviewers are satisfactorily addressed.

Reviewer #3 (Remarks to the Author):

In my opinion, the authors have done much effort to successfully improve their manuscript both by additional experimental results and discussion/explanations. They have replied adequately to all reviewers' comments (I looked at the replies to all three reviewers) and took them into account for revising the manuscript. Taking into account the already positive view of reviewers 2 and 3, I recommend the publication of this revised version in Nature Communications. Within their responses the authors mention new BMP experiments they have performed "in the meantime". If these results are not too long, it would be interesting to show them in the manuscript. Otherwise (if this would extend the present manuscript too much) I would agree with the authors to publish those results in subsequent articles. Then they could still mention them quickly and also that current ongoing work is investigating these issues.

Point-by-point response to the Reviewer's Comments

REVIEWERS' COMMENTS:

Reviewer #1 (Remarks to the Author):

The comments from the reviewers are satisfactorily addressed.

>> Thank you.

Reviewer #3 (Remarks to the Author):

In my opinion, the authors have done much effort to successfully improve their manuscript both by additional experimental results and discussion/explanations. They have replied adequately to all reviewers' comments (I looked at the replies to all three reviewers) and took them into account for revising the manuscript. Taking into account the already positive view of reviewers 2 and 3, I recommend the publication of this revised version in Nature Communications. Within their responses the authors mention new BMP experiments they have performed "in the meantime". If these results are not too long, it would be interesting to show them in the manuscript. Otherwise (if this would extend the present manuscript too much) I would agree with the authors to publish those results in subsequent articles. Then they could still mention them quickly and also that current ongoing work is investigating these issues.

>> Thank you. In the revised manuscript we mention current ongoing work.